# CRISPR-based gene drives generate super-Mendelian inheritance in the disease vector *Culex quinquefasciatus*

Tim Harvey-Samuel [1,7], Xuechun Feng [2,3,7] ✉, Emily M. Okamoto [2], Deepak-Kumar Purusothaman [1,5], Philip T. Leftwich [4], Luke Alphey [1,6] ✉ & Valentino M. Gantz [2] ✉

*Culex* mosquitoes pose a significant public health threat as vectors for a variety of diseases including West Nile virus and lymphatic filariasis, and transmit pathogens threatening livestock, companion animals, and endangered birds. Rampant insecticide resistance makes controlling these mosquitoes challenging and necessitates the development of new control strategies. Gene drive technologies have made significant progress in other mosquito species, although similar advances have been lagging in *Culex*. Here we test a CRISPR-based homing gene drive for *Culex quinquefasciatus*, and show that the inheritance of two split-gene-drive transgenes, targeting different loci, are biased in the presence of a Cas9-expressing transgene although with modest efficiencies. Our findings extend the list of disease vectors where engineered homing gene drives have been demonstrated to include *Culex* alongside *Anopheles* and *Aedes*, and pave the way for future development of these technologies to control *Culex* mosquitoes.

The southern house mosquito *Culex quinquefasciatus* is a cosmopolitan vector of zoonotic arboviruses including West Nile virus (WNV), St. Louis encephalitis, and Western Equine encephalitis, and is the primary vector of *Wuchereria bancrofti*, the causal agent of lymphatic filariasis[1,2]. These neglected tropical diseases are of serious global concern, with WNV alone being the most commonly acquired arboviral disease in the continental USA and responsible for the three largest arboviral neuroinvasive disease outbreaks recorded there[3]. Beyond its importance to human health, *Cx. quinquefasciatus* and the diseases it vectors are significant and continuing threats to insular avian communities across the tropics and subtropics, including many unique and endangered species[4]. As an example, its invasion of Hawai'i in the 1800s, alongside the introduction of avian malaria and avian pox, led to the partial collapse of forest bird communities on that archipelago[5].

*Cx. quinquefasciatus* is an important 'bridge' vector for these and other zoonoses as it feeds readily on a variety of mammals (including humans) as well as other animals (significantly, bird species)[6]. This flexibility in host selection, as well as a high level of environmental adaptability, has led to this mosquito invading multiple environments over the last ~200 years—often causing novel disease transmission cycles upon reaching new areas[7]. Conventional methods for controlling *Cx. quinquefasciatus* are insufficient to prevent its further spread or continued outbreaks of its associated diseases[4]. With climate change predicted to increase the disease burden of this mosquito—including range expansion into western Europe[8], development of novel, effective, and environmentally friendly control technologies is required.

Engineered 'gene drives' represent one such technology. These are heritable genetic elements, transmitted via mating, which can

[1]Arthropod Genetics Group, The Pirbright Institute, Woking GU24 0NF, UK. [2]Section of Cell and Developmental Biology, University of California San Diego, La Jolla, CA 92093, USA. [3] Institute of Infectious Diseases, Shenzhen Bay Laboratory, Guangdong 518106 Shenzhen, China. [4]School of Biological Sciences, University of East Anglia, Norwich Research Park, Norwich NR4 7TJ, UK. [5]Present address: MRC-University of Glasgow Centre for Virus Research, University of Glasgow, Glasgow G12 8QQ, UK. [6]Present address: Biology Department, University of York, York YO10 5DD, UK. [7]These authors contributed equally: Tim Harvey-Samuel, Xuechun Feng. ✉e-mail: fengxc@szbl.ac.cn; luke.alphey@york.ac.uk; vgantz@ucsd.edu

spread autonomously from low to high frequency in a population, despite not conferring a fitness advantage[9–11]. Gene drives can be engineered to target genes regulating female or male fertility, thereby reducing the target population density (known as population suppression), or to cargo a 'disease-refractory transgene' and reduce the capabilities of a pest population to vector pathogens (known as population modification)[12–14]. Theoretically, gene drives represent a more species-specific and cost-effective approach than currently deployed control technologies due to their autonomous behavior once released into a target population, leading to significant research interest in deploying them against human-disease vectors, agricultural pests, and invasive species[15,16].

The most developed gene drive platform is the 'homing drive'[17], based on CRISPR/Cas9 technology[18]. CRISPR/Cas9 includes a targetable nuclease (Cas9) which binds DNA causing precise double-stranded breaks (DSBs), and a 'guide RNA' (gRNA) which guides the Cas9 nuclease to DNA sequences complementary to its sequence. In a traditional homing drive, transgenes expressing Cas9 and a gRNA are integrated into a genome at the site specified by that gRNA. In gene drive heterozygotes, the wildtype chromosome is cleaved by Cas9 and the gene drive chromosome can then be used as a template for homology-directed repair (HDR), copying the entire gene drive transgene to the wildtype chromosome (known as a 'homing' reaction). If this homing reaction occurs in the germline of the target organism it will lead to biased (>50%) inheritance of the gene drive in the next generation (super-mendelian inheritance) and, depending on other parameters such as associated fitness costs, an increase in the population drive allele frequency[18,19].

While homing drive technology has been demonstrated in the yellow fever mosquito *Aedes aegypti* as well as various malarial mosquitoes (*Anopheles gambiae, Anopheles stephensi*)[14,20–25], development of gene drives in *Culex* mosquitoes has not yet taken place. Having recently developed CRISPR tools necessary for site-specific integration of transgenes into the *Cx. quinquefasciatus* genome[26–29], we set out to assess whether the mechanism underlying homing drives could be demonstrated in this important pest species. Using a 'split-drive' design with our previously developed *vasa*-Cas9 source integrated at the *cardinal* locus[26] (a separate, non-homing locus) and gRNAs ('homing elements') integrated into either the *white* or *kynurenine 3-monooxygenase* (*kmo*) loci (both genes involved in eye pigment synthesis pathways and thus providing a readily scorable knockout phenotype[27,29]) we observed significant homing at both locations. The success of this initial proof-of-principle test, at multiple loci, is promising for the further development of gene drives for population manipulation in *Culex* mosquitoes.

## Results

### Generation of a gRNA-only homing element mosquito line targeting white

To test whether a CRISPR-based homing gene drive is feasible in *Cx. quinquefasciatus*, we employed a split, gRNA-only gene-drive strategy that takes advantage of a *vasa*-Cas9 line that we previously built[26] as a source of Cas9 protein. As this Cas9 transgene was inserted at the *cardinal* locus, we decided to utilize another gene causing a visible phenotype, the *white* gene (CPIJ005542), as a homing target. Both *white* and *cardinal* are located on chromosome I of *Cx. quinquefasciatus*, but are genetically unlinked, being located on opposite chromosomal arms. Additionally, we note that *white* is known to be tightly linked to the sex-determining region in both *Culex* and *Aedes* (hereon, we use 'M' to represent the allele carrying the Male-determining sequences and 'm' to represent the corresponding locus lacking the male-determining ability; therefore males are indicated as M/m, and females are indicated as m/m) (Fig. 1a)[30,31]. The *vasa*-Cas9 transgene used to test the split gene-drive arrangement was previously generated by integrating a Cas9 transgene expressed under the

control of regulatory sequences from the *Cx. quinquefasciatus vasa* gene, and marked by a *DsRed* gene controlled by the *Opie2* promoter (Fig. 1b). We then built the second component, the 'homing' gRNA-only drive by inserting it into exon 5 of the *white* locus at the exact cut site specified by the *white*-gRNA6, a construct comprising: (1) the *white*-gRNA6 under the control of the *Cx. quinquefasciatus* U6:1 promoter, which carries the "Loop" modified scaffold version for increased efficiency;[26,32] (2) an *eGFP* fluorescent marker under the control of the *Hr5IE1* promoter to track the gRNA transgene; and (3) an in-frame re-coded portion of the *white* gene that would produce a functional *white* protein thereby restoring the endogenous gene's activity (Fig. 1c). The targeted transgene delivery was achieved by adding, to the plasmid, two ~1 kb homology arms (HAs) matching the genomic sequence of the *white* locus abutting the *white*-gRNA6 cutting site (Fig. 1c).

To generate transformant mosquitoes carrying this *white*-gRNA6-only drive construct, the plasmid depicted in Fig. 1c was injected into eggs collected from the *vasa*-Cas9 line. The surviving G0 mosquitoes were then separated into male and female pools and outcrossed to our wildtype line (see methods for details). The resulting 693 G1 progeny were phenotypically screened for the presence of the eGFP marker, and 1 transformant male was recovered (Supplementary Data 1). This eGFP+/DsRed+ male, carrying both the *white*-integrated gRNA-drive and *cardinal*-integrated *vasa*-Cas9 transgenes, was then mated to 20 virgin wildtype females to establish a transgenic line (Supplementary Data 2). From this cross, nearly all of the recovered male offspring displayed eGFP fluorescence, suggesting that the *white*-gRNA6 transgene had likely inserted at the *white* locus in tight linkage with the sex-determining region (Fig. 1a and Supplementary Data 2), and specifically, linked to the M allele. From the same cross, we also recovered a single eGFP+ female, providing a first indication that the *white*-gRNA6 element was potentially capable of homing onto the opposing, m-allele-carrying chromosome in the presence of the Cas9 transgene (Supplementary Data 2), although, from these results, we cannot rule out a recombination event occurring between the two loci. This single eGFP+ female was then crossed to 8 eGFP+ males in order to isolate a second m-linked *white*-gRNA6 transgenic line (Supplementary Data 2). The insertion of the transgene was confirmed by PCR amplification and Sanger sequencing, indicating the integration of all the functional portions of transgenes along with the backbone (Supplementary Fig. 2). Since the two insertion junctions were seamless with the surrounding genomic flanking region, as confirmed by Sanger sequencing, the presence of the backbone indicates a tandem integration of the construct, which, if limited in the amount of repeats, should not impact its functionality and ability to home onto the opposing chromosome.

### Assessing gene drive at the *white* locus using an inheritance-bias approach

In our transgenic recovery strategy described above, we were able to obtain two separate lines: the first carrying the *white*-gRNA6 homing element tightly linked to the M-locus and homozygous for the *vasa*-Cas9 transgene, and a second line carrying the *white*-gRNA6 homing element tightly linked to either the M-locus or the m-locus, in an otherwise wildtype background. These two lines allowed us to assess whether local sequence differences around the sex locus impair chromosomal pairing and efficient homing onto the homologous chromosome. We utilized each of these lines to assess the homing of the *white*-gRNA6 drive and evaluate the conversion efficiency in either males, with the *white*-gRNA6 insertion homing from an M-linked to an m-linked locus (M-to-m homing), or in females with the *white*-gRNA6 insertion homing from an m-linked to another m-linked locus (m-to-m homing).

To assess M-to-m homing, we mated 10 G0 males from the *white*-gRNA6; *vasa*-Cas9 line to 10 wildtype G0 females. From their progeny, we isolated trans-heterozygous G1 males that carried both the *vasa*-

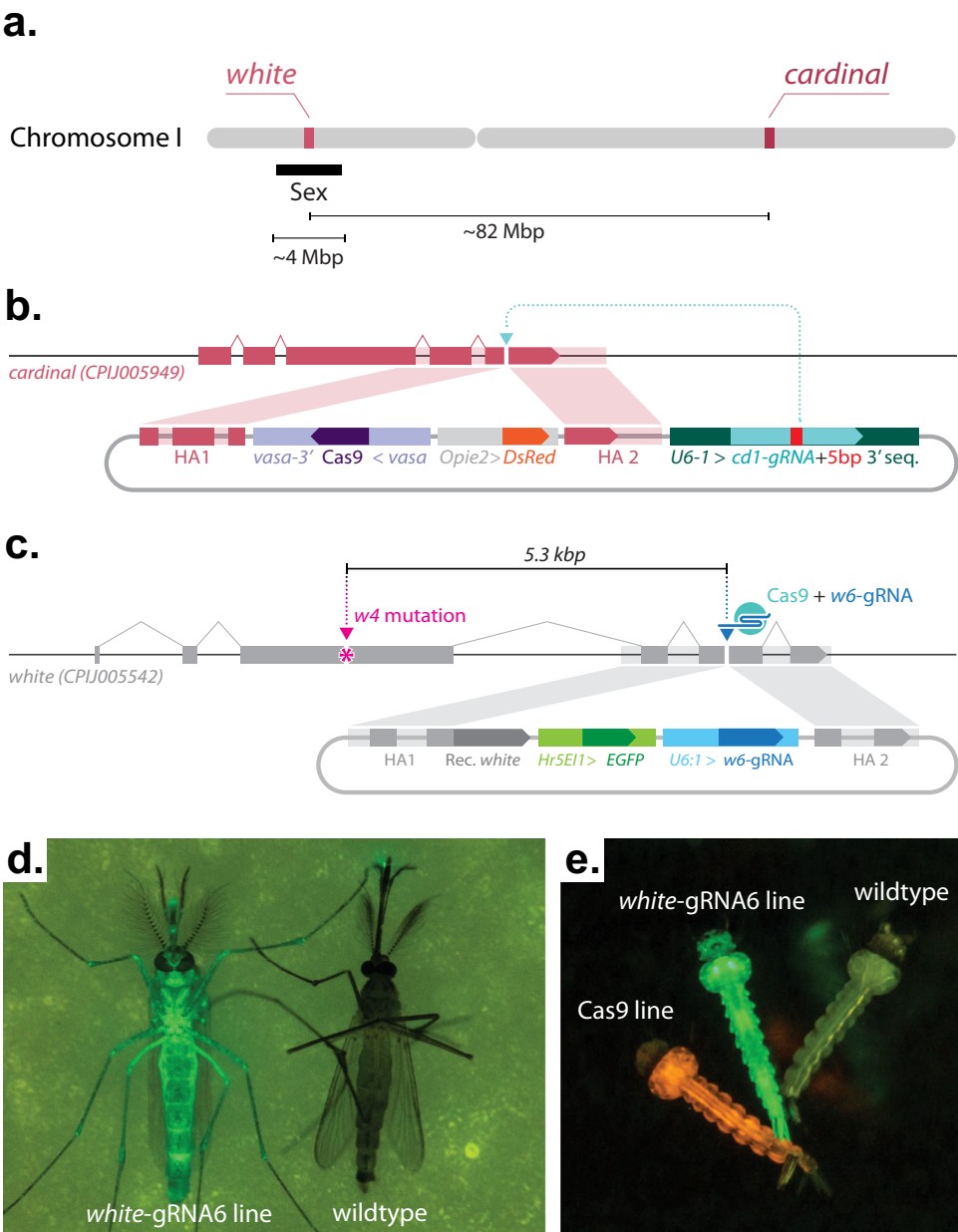

**Fig. 1 | Constructs built and transgenic mosquitoes generated to test a split-drive system in *Culex quinquefasciatus* targeting the *white* locus. a** Schematic outline of targeted gene locations in *Cx. quinquefasciatus* chromosome I. The approximate distances between different loci are indicated. The approximate location of the sex-determining locus is indicated with a black bar. Drawings are not to scale. **b** Graphical representation of the plasmid construct and transgene integration strategy used to build the *vasa*-Cas9 line in our previous study[26]. **c** The gRNA-eGFP construct and transgene integration strategy for building the *white*-gRNA6 homing element line. **b** Pink and **c** gray shading highlight homology arms (HAs) correspondence to the genome sequences used for site-directed integration. The *w4*- mutation marked with a magenta asterisk (*) sits a distance of 5.3 kb from the *white-gRNA6* target site located on exon 5. Graphs are not to scale. **d** Photo of a transgenic adult mosquito expressing eGFP (left) compared to a wildtype mosquito (right). **e** Photo of three larvae: left, *vasa*-Cas9 line expressing DsRed; middle, *white*-gRNA6 line expressing eGFP; right, a wildtype larvae without fluorophore expression.

Cas9 and *white*-gRNA6 transgenes so as to ensure that the only *white*-gRNA6 element present was M-linked. We then mated these trans-heterozygous males to virgin wildtype females in single pairs to obtain and analyze each of their G2 progeny independently (Fig. 2a). Separately, to assess m-to-m homing, we mated 10 *white*-gRNA6 G0 females to 10 *vasa*-Cas9 G0 males, and from their offspring we isolated G1 trans-heterozygous females carrying both transgenes and crossed them to wildtype males in single pairs to obtain and analyze their G2 progeny (Fig. 2b). For both crossing strategies, the phenotypic analysis of the fluorescence ratios in the G2 generation allowed us to evaluate the homing efficiency of the *white*-gRNA6 element (Fig. 2a, b).

For the M-to-m homing assessment, the G2 progeny of 7 G1 crosses were scored and an average inheritance of 51.2% [95% CI: 45.2%–57.1%] was observed for the *white*-gRNA6 transgene, showing no significant difference (GLMM: $\beta = 0.05 \pm 0.12$, Wald statistic $z = 0.4$, $p = 0.67$) from the expected 50% inheritance ratio of Mendelian inheritance (Fig. 2c, Supplementary Data 3). For the m-to-m homing assessment, the G2 progeny derived from 19 trans-heterozygous female crosses were evaluated and a modest but significant inheritance bias of 54.8% [95% CI: 51.6%–57.9%] was observed (GLMM: $\beta = 0.19 \pm 0.07$, $z = 2.75$, $p = 0.006$) (Fig. 2d, Supplementary Data 3). When analyzing the inheritance of the *vasa*-Cas9 transgene, we found

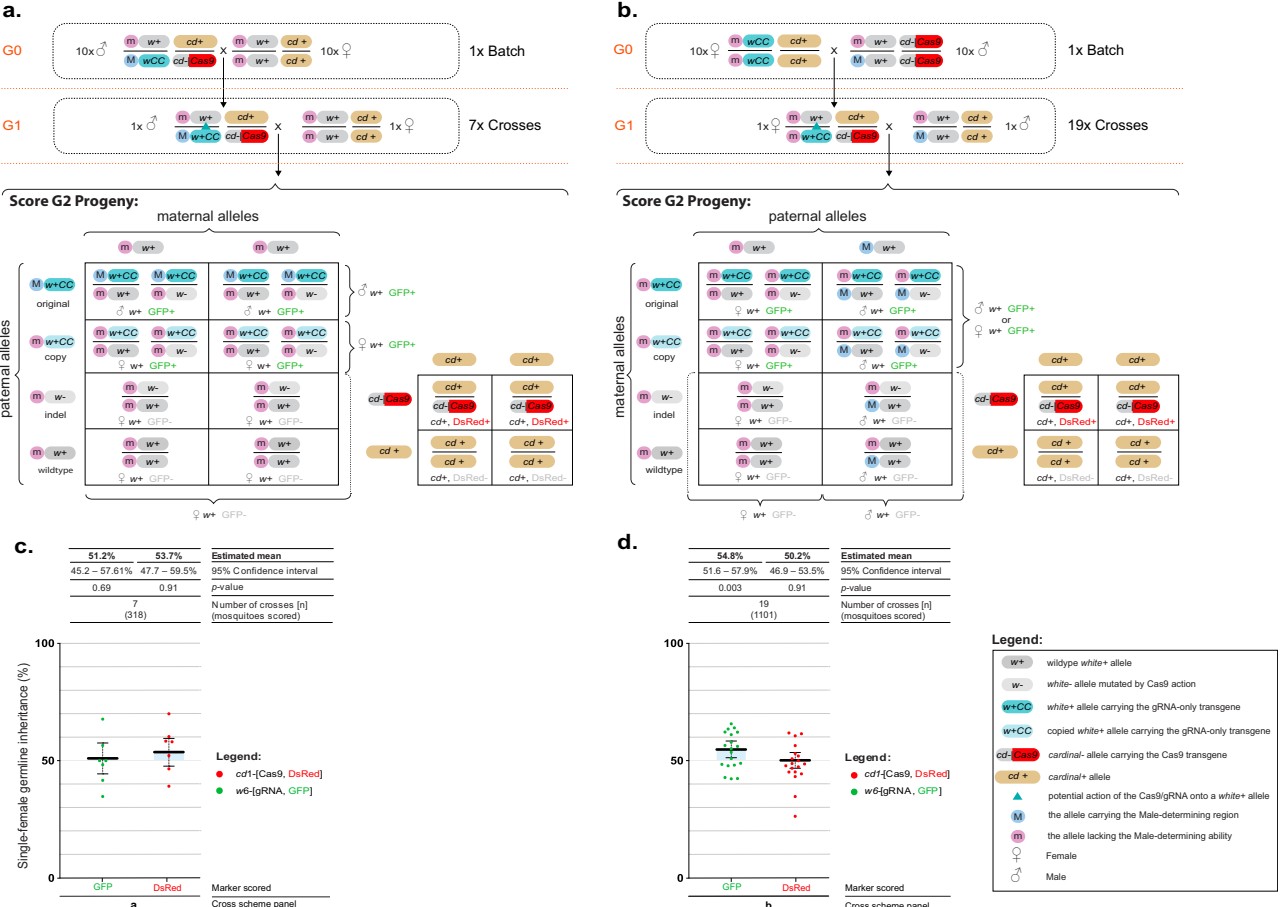

**Fig. 2 | A split gene drive targeting the *white* gene in *Culex quinquefasciatus*.**
**a** Genetic cross scheme for the M-to-m homing analysis: transgenic males carrying both genetic elements were crossed to multiple virgin females. Their trans-heterozygous G1 male progeny were single-pair crossed with wildtype virgin females to obtain and score their G2 offspring. **b** Genetic cross scheme of the m-to-m homing analysis: females carrying the *white*-gRNA6 transgene were crossed to *vasa*-Cas9 males. From their progeny, trans-heterozygous G1 females were selected and single-pair crossed with wildtype males to obtain and score their G2 progeny. **a**, **b** Below each genetic cross is a Punnett square representation of the expected genotypes. The *white* allele carrying the *white*-gRNA-eGFP transgene is marked with

cyan; the copied allele is marked with light blue. Potential allelic conversion events (homing) are indicated by blue inverted triangles. **c** A graph summarizing the inheritance rates of the *white*-gRNA6/eGFP and *vasa*-Cas9/DsRed transgenes in the G2 progeny derived from G1 male germlines in the M-to-m cross strategy. **d** The inheritance rates of *white*-gRNA6/eGFP and *vasa*-cas9/DsRed transgenes in the G2 progeny derived from G1 female germlines in the m-to-m cross strategy. Estimated means and 95% confidence intervals were calculated by a generalized linear mixed model, with a binomial ('logit' link) error distribution, and are presented above the graphs for each data set. Raw phenotypic scoring is provided in Supplementary Data 3.

no evidence for inheritance bias of this transgene (Males: 53.7% [95% CI: 47.7%–59.5%]; Females: 50.2% [95% CI: 46.9%–53.5%]). In summary, we seem to observe a slight inheritance bias only when the split drives homes in the m-to-m condition, and not in the M-to-m, suggesting that homing could be potentially impaired at this locus when homing occurs between chromosomes with slight differences in the local sequences. Additionally, since it seems that the homing process may occur at very low levels (Fig. 2d), homing events may also be happening in the M-to-m condition, which might be obscured with the smaller sample size and the limited resolution of this assay (Fig. 2c).

**Assessing gene drive at the *white* locus using a marked chromosome approach**
During our initial experiment, we noticed a modest yet significant bias in the inheritance of the *white*-gRNA6 transgene. To better understand the underlying chromosomal conversion events, we designed a follow-up experiment that would allow us to differentiate between homing events occurring through HDR, and the biased inheritance of the whole gene drive-bearing chromosome.

In order to investigate the inheritance of the *white*-gRNA6 transgene, we designed a more sensitive assay. This assay involved using a

marked-chromosome approach by using a specific mutation closely linked to the homing site on the receiver "m" chromosome. This mutation is not present on the donor, *white*-gRNA6-element-containing "M" chromosome (Fig. 3a). If a successful homing event occurred, it would be identified by the linkage of the *white*-gRNA6 element with this unique marker mutation on the receiver "m" chromosome. This marker mutation, hereon referred to as *w4-*, was repurposed from a *white* mutant line that we previously built[27], which carries a disruption of the *white* coding sequence under the action of the *white*-gRNA4, targeting exon 3 (*w4* mutation, Figs. 1c and 3a), and sitting at a 5.6 kbp distance from the *white*-gRNA6 element insertion site. Contrary to the sex-determining locus, which while being tightly linked with *white* allows for recombination in between the markers, the *w4-* mutation is too close to the *white*-gRNA6 insertion site to generate meaningful recombination in the intervening sequence.

To conduct the analysis we first crossed homozygous *vasa*-Cas9 females with homozygous *w4-/w4-* mutant males to obtain trans-heterozygous *w4+/w4-; cd-,vasa*-Cas9+/cd+ offspring (Fig. 3b). Subsequently, we intercrossed these individuals to generate and isolate progeny with a DsRed+/*w4-* phenotype, which would be homozygous for the *w4-* mutation (white eyes) and carrying at least one copy of the *vasa*-

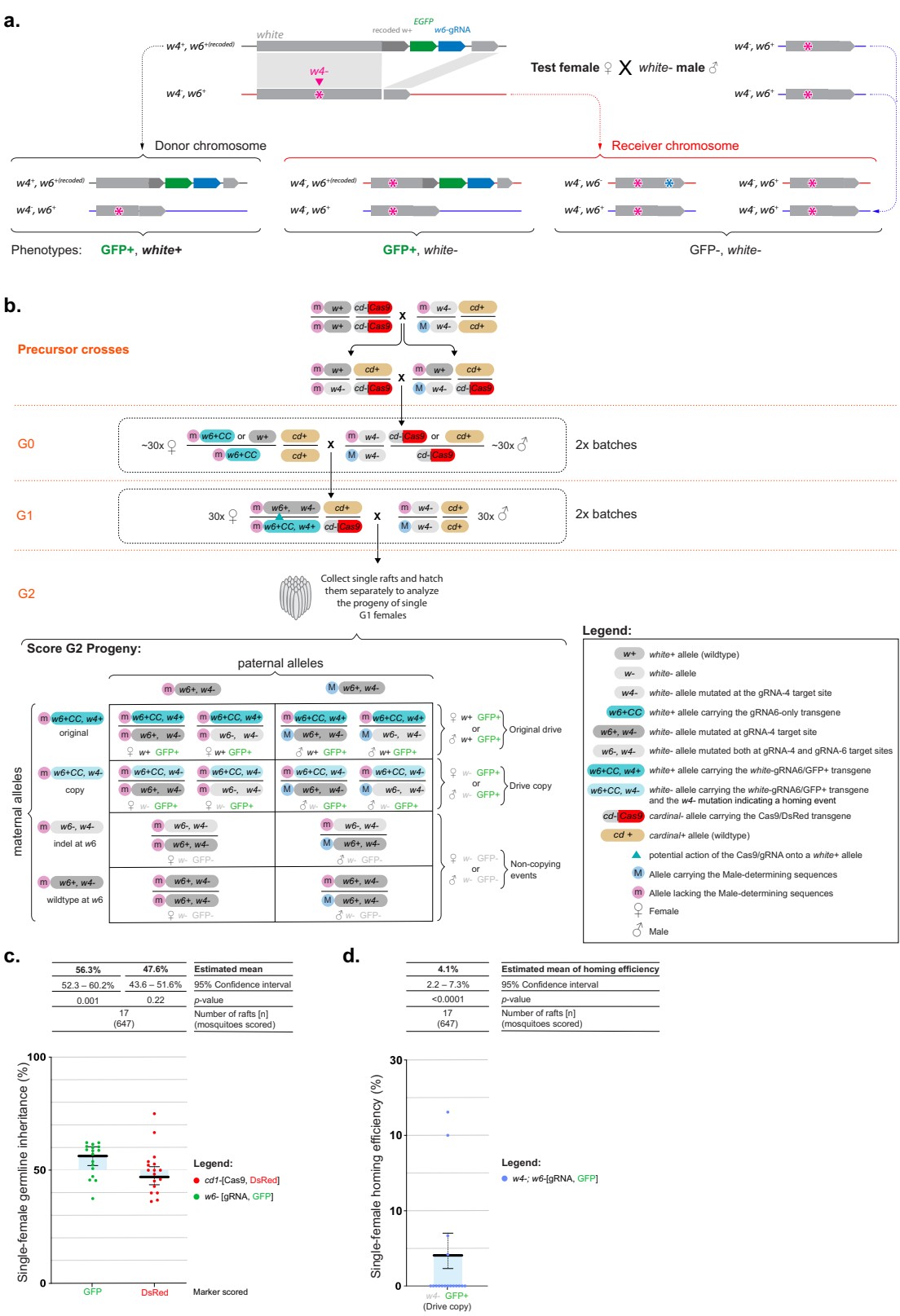

Cas9 transgene (DsRed+) (Fig. 3b). To set up the marked-chromosomes homing analysis, we took 30 of these DsRed+/*w4-* G0 males and crossed them in a pool with 30 G0 virgin females containing either one or two copies of the *white*-gRNA6 transgene (Fig. 3b). From the resulting offspring we chose trans-heterozygous G1 virgin females carrying both the *vasa*-Cas9 (DsRed) and the *white*-gRNA6 (eGFP) transgenes, along with the *w4-* mutation in heterozygotes (*w4+/w4-*). These G1 females were

then pool-mated to *w4-/w4-* males, and the resulting G2 offspring was collected at the egg-raft stage and allowed to hatch in separate containers. This setup enabled us to assess the inheritance of the *white*-gRNA6 in the germline of each G1 females by examining the eye phenotype and marker presence of their G2 offspring (Fig. 3b).

Since the *w4-* precedes the *white*-gRNA6 insertion site on the *white* coding sequence (Fig. 1c), by scoring the eye color of the G2

**Fig. 3 | A marked chromosome split gene drive in *Culex quinquefasciatus*.**
**a** Graphical representation of the experimental design, which allows to track donor and receiver chromosomes. **b** Genetic cross scheme used to test homing efficiencies using a marked chromosome approach. Precursor crosses: *w4-* homozygous males were crossed with *cd-,vasa-*Cas9 homozygous females to generate *w4-/w4+*; *cd-,vasa-*Cas9*/cd+* trans-heterozygote offspring, which was then intercrossed to isolate G0 *w4-/w4-*;DsRed+ males carrying at least one copy of the *cd-,vasa-*Cas9 allele. These G0 males were then mated to *white*-gRNA6-eGFP females. From their female progeny we isolated G1 test animals carrying both DsRed+ and eGFP+ fluorescent markers, which would have the *w4+,white*-gRNA6*/w4-,w6+*; *cd-,vasa-*Cas9*/cd+* genotype. The test G1 females were then crossed in a pool with *w4-/w4-* males. The females were allowed to lay eggs, at which point single rafts

were collected, and hatched separately to perform a phenotypical analysis of each independent germline. Potential homing events are indicated by blue triangles in the cross scheme. Below the genetic cross is a Punnett square representation of the expected genotypes with the *white*-gRNA-eGFP transgene labeled in either dark cyan (original) or light cyan (copy). **c** Graph reporting the inheritance rates observed for the transgenes in the G2 progeny derived from G1 females. **d** Homing efficiency rate calculated using the receiver chromosomes only, in the marked-chromosome strategy. Estimated means and 95% confidence intervals were calculated by a generalized linear mixed model, with a binomial ('logit' link) error distribution, and are presented above the graphs for each data set. Raw phenotypic scoring is provided in Supplementary Data 3.

offspring we were able to distinguish the original *white*-gRNA6 which had dark eyes (*w4+,white*-gRNA6*/w4-,w6+*, where *w6+* indicates wild-type sequence for the *white* coding sequence at the *white*-gRNA6 location), from the 'homed' copies which instead displayed a white-eye phenotype (*w4-,white*-gRNA6*/w4-,w6+*) (Fig. 3b). Consistent with our previous experiment (Fig. 2d), in the G2s, we observed an average inheritance rate for the eGFP+ marker of 56.3% [95% CI: 52.3 –60.2%], significantly above the null hypothesis of Mendelian inheritance (GLMM: $\beta = 0.25 \pm 0.08$, $z = 3.18$, $p = 0.001$) (Fig. 3c). Furthemore, by implementing this multi-step procedure involving the *w4-* allele and the eGFP marker, we could assess both homing events and biased inheritance of the donor chromosome, providing us with a comprehensive understanding of transgene inheritance as this locus. Homing events occurred in 4 out of 15 of the analyzed rafts, and in 4.1% (95% CI: 2.2–7.3%) of the total receiver chromosomes we observed homing, which caused a mean bias in inheritance of 1.9% [95% CI: 1.0%–3.3%] (Fig. 3d). Separately, due to our ability of tracking donor and receiver chromosomes, we also found evidence of a significant bias in the inheritance of the donor chromosome at a mean rate of 4.4% [95% CI: 0.4–8.4%], higher than predicted by mendelian inheritance. Taken together, homing-generated bias and increased donor-chromosome inheritance account for the total bias in the eGFP+ marker observed (Supplementary Data 3).

### Assessing gene drive at the *kmo* locus using an inheritance bias approach

Previous work on split-drive systems in *Ae. aegypti* identified a strong influence of the target site on homing efficiency, even when using the same Cas9 expressing line (e.g., comparing the inheritance rates of homing elements integrated into the *kmo*[33] and *white*[25] genes in *Aedes aegypti* when paired with the same *SDS3*-Cas9 line). To assess whether the homing rate might be higher at a target site different to *white*, we paired the *vasa*-Cas9 line with our previously established gRNA homing element inserted into exon 5 of the *kynurenine 3-monooxygenase* (*kmo*) gene (*kmo*-gRNA line)[29,34]. Unlike the *white*-gRNA6 line used above, this homing element did not contain a *kmo* rescue fragment and thus represents a null allele for the gene function, resulting in eye lacking pigment. In order to test homing of the *kmo*-gRNA drive, male and female pupae from the homozygous *vasa*-Cas9 line were sexed and reciprocally mated *en masse* to heterozygous individuals from the *kmo*-gRNA line (the G0 generation). Progeny from each of the two crosses were screened as pupae for the presence of the two transgenic markers as well as for any eye color mosaic phenotypes resulting from Cas9-based disruption of *kmo* (Supplementary Fig. 4). Unlike for the *white*-gRNA6 experiment, where the two transgenes were differentiated using different fluorescent markers, in this analysis we used transgenic lines both marked with DsRed; fortunately we were able to identify differences in the DsRed expression pattern driven by either the *Hr5IE1* or the *Opie2* promoters that allowed us to clearly identify animals carrying each of the transgenes (Supplementary Fig. 3). It was noted in both crosses that all trans-heterozygous pupae (those that had inherited both transgenes) displayed a strong *kmo* mosaic

phenotype, while no such phenotype was observed in progeny inheriting only the *vasa*-Cas9 transgene (Supplementary Fig. 4). Transheterozygous G1 animals from each cross were sexed and independently crossed *en masse* to wildtype individuals of the opposite sex (giving four pooled-cross cages). One week after crossing, cages were blood-fed and egg rafts were collected and allowed to hatch and recorded individually. Larval progeny from each egg raft were screened for the presence of the two transgenes (the G2 generation). For each egg raft replicate, *kmo*-gRNA and wildtype G2 progeny were isolated and allowed to continue developing till pupation, at which point they were screened for the eye color phenotype, indicative of *kmo* activity.

Overall, we observed a *kmo*-gRNA inheritance rate of 57.5% [95% CI: 55.5–59.6%] equating to approximately 15% of wildtype alleles being converted (homed) to *kmo*-gRNA alleles in the G1 germline, a highly significant difference from the null hypothesis of Mendelian inheritance (GLMM: $\beta = 0.24 \pm 0.03$, $z = 7$, $p < 0.001$). No significant effect was observed regarding the Cas9-bearing sex of the G0 generation; however, when this treatment was collapsed, sex of the transheterozygous parent in the G1 cross had a marginally significant effect on *kmo*-gRNA inheritance (progeny of female $G_1$ transheterozygotes = 55.0% [95% CI: 53.2–56.7%] for *kmo*-gRNA inheritance, progeny of male $G_1$ transheterozygotes = 60.0% [95% CI: 56.3–63.7%] for *kmo*-gRNA inheritance (Fig. 4). Akaiake information criteria (AIC) selection confirmed this was the minimal significant model.

### Transgenerational deposition of Cas9/*kmo* gRNA ribonucleoprotein (RNP)

By screening the *kmo*-gRNA-expressing and wildtype G2 progeny of the above experiment for their *kmo* phenotype, we were able to assess whether RNP was being transferred either maternally or paternally between generations. In these cohorts, we observed a substantial number of mosaic-eye color individuals (Fig. 5), indicating transgenerational deposition of Cas9 and the *kmo*-targeting gRNA from the G1 generation into G2 embryos. These mosaics occurred along a spectrum of severity−from 'faint' to 'strong', alongside fully white-eyed individuals (Fig. 5a–e). In cohorts where these mosaics occurred, they were generally observed in less than 50% of individuals screened, with the highest proportion of mosaics being grouped into the 'faint' category and fewer 'strong' category or completely white-eyed individuals observed. While substantial levels of mosaicism were observed in the progeny of G1 transheterozygous females, three out of the four G2 cohorts deriving from G1 transheterozygous males displayed no mosaicism or white eyes at all, with the fourth showing only two faint mosaic individuals (c. 5% of individuals in that cohort, Fig. 5l).

## Discussion

In this study, we built and successfully tested an engineered CRISPR-based homing gene drives in *Culex quinquefasciatus*. We observed biased inheritance at two independent loci, *white* (CPIJ005542) and *kynurenine 3-monooxygenase* (CPIJ017147), and provided robust evidence for the occurrence of Cas9-induced homing reactions in a *Culex*

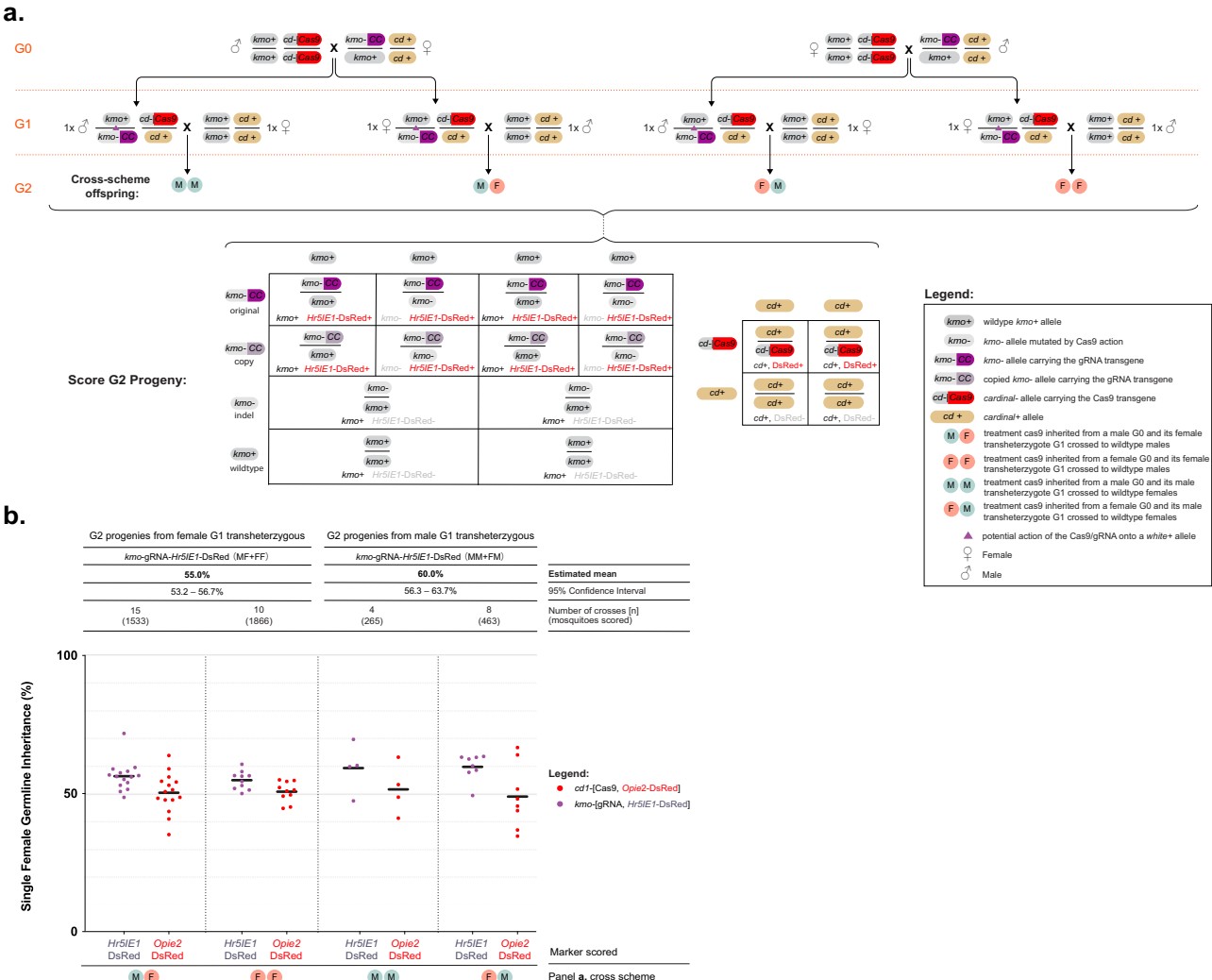

**Fig. 4 | A split gene drive targeting the *kmo* gene in *Culex quinquefasciatus*.**
**a** Genetic cross scheme for the experiment. G0 individuals carrying the *kmo*-gRNA transgene were reciprocally mated *en masse* to individuals carrying the *vasa*-Cas9 transgene. Male and female transheterozygote progeny from each of these two crosses were mated *en masse* to wildtype of the opposite sex (G1). Inheritance of *vasa*-Cas9 and *kmo*-gRNA transgenes scored in their progeny (G2). The different phenotypes of G2 individuals are listed. Potential allelic conversion events (homing) are indicated by blue inverted triangles in the cross scheme.

**b** The inheritance rates of *kmo*-gRNA (marked by *Hr5IE1*-DsRed) and *vasa*-Cas9 (marked by *Opie2*-DsRed) transgenes in the G2 progeny. Estimated means and 95% confidence intervals were calculated by a generalized linear mixed model, with a binomial ('logit' link) error distribution, and are presented above the graphs for each data set. Raw data points are given in the graph whereas modeled data is provided in the main text. Raw phenotypic scoring is provided in Supplementary Data 3.

mosquito. The estimates for biased inheritance at both loci were relatively modest (~10% at the *kmo* locus for male G1 trans-heterozygotes and ~6% at the *white* locus for female G1 trans-hetero-zygotes), yet support the future development of gene drive technologies in these mosquitoes.

Direct comparison between results at these two loci is complicated by multiple factors that may have impacted the homing rates observed. Differences in homing element design, such as different gRNA efficiencies, different PolIII promoters used for gRNA expression, the modified gRNA scaffold improving gRNA efficiency employed in the *white*-gRNA6[26], and the tandem insertion of the *white*-gRNA6 construct, which included the plasmid backbone, could have affected the homing process, making it complicated to compare the constructs used at the two loci (Supplementary Fig. 2d). In addition, the genetic linkage of the *white* locus to the sex-determination locus (M factor, which may have degenerate sequence homology) may suppress HDR locally. This, could have negatively impacted the homing-based biased inheritance of the *white*-gRNA6 construct, similar to previous

observations in *Ae. aegypti*[24,25]. We employed a marked-chromosome strategy to identify homing events at the *white* locus that confirmed this is occurring and that an additional contributing effect also enhances the biased inheritance, likely a meiotic drive component destroying some of the receiver chromosomes. While we could confirm these two effects for *white*, our *kmo* experimental setup did not allow disentangling these two effects. However, we acknowledge that this may also contribute to the biased *kmo* inheritance observed.

When contrasting these results to other studies, the most appropriate mosquito for comparison is *Ae. aegypti*—the closest species to *Cx. quinquefasciatus* in which homing drives have been engineered and a co-member of the Culicinae subfamily. In the case of the *kmo*-gRNA element, mean homing levels observed (c.10% for females and c.18% for males) were within the error ranges estimated for some previously reported split-drive systems in *Ae. aegypti*[24,25], with significant levels of homing (e.g. *exu*-Cas9; U6c-gRNA / U6D-gRNA females = 12% / 6%, *trunk*-Cas9; U6b-gRNA females = 10%). Similar to what we observed at the *white* locus, previous studies in *Ae. aegypti*[24,25]

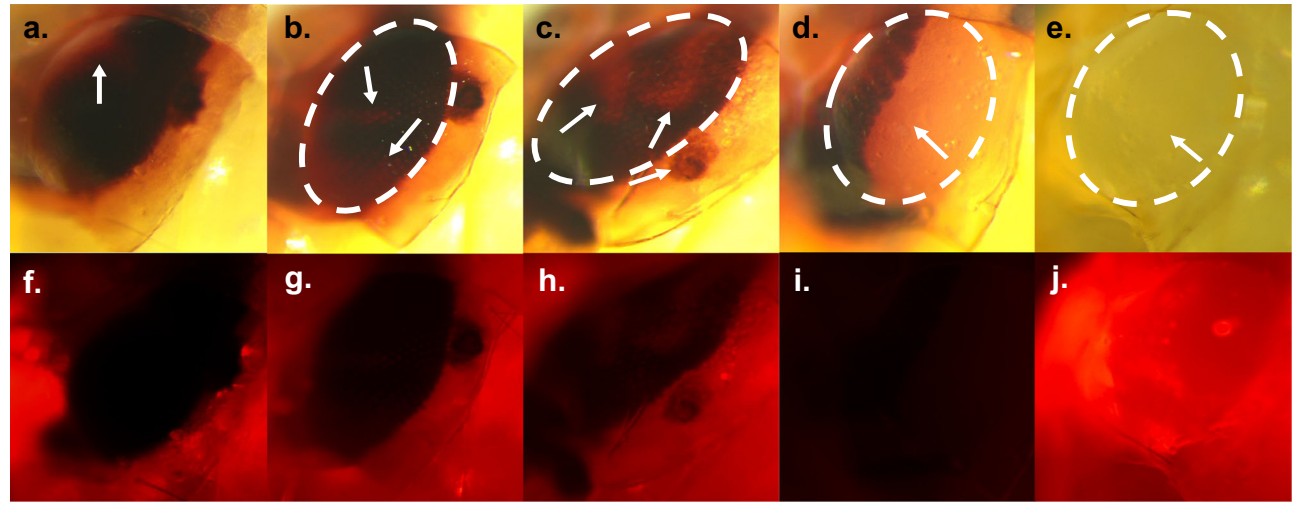

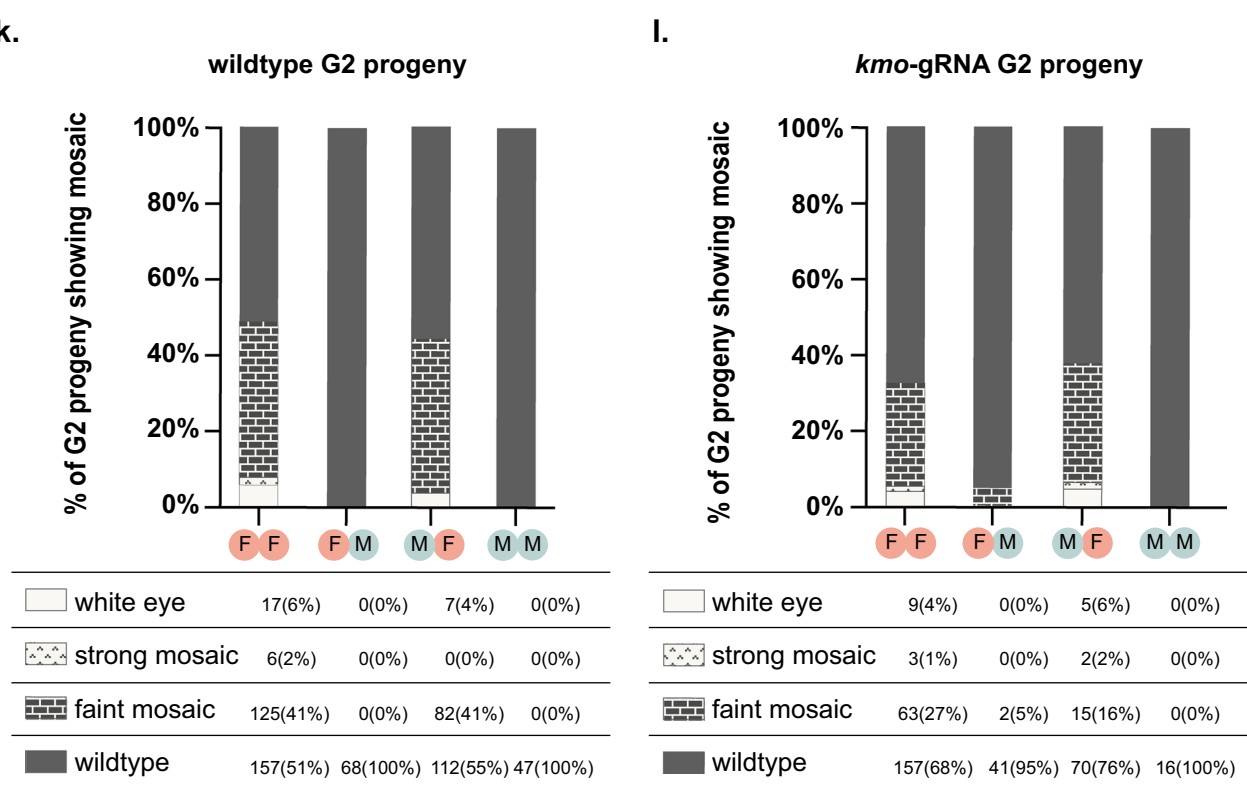

**Fig. 5 | Deposited RNP leads to *kmo* mosaicism in G2 progeny. a–j** Representative photographs of the different phenotypic classes covering the spectrum of phenotypes caused by *kmo* gene disruption. (**a–c**, **f–h**) 'faint mosaics', (**d**, **i**) 'strong mosaics', and (**e**, **j**) white eyes. Individual in **d**, **l** is a wildtype G2 progeny. Other individuals are *kmo*-gRNA heterozygote G2 progeny. Approximate outline of the eye tissue in each of the upper panels is shown by the white dotted oval. Patches of mosaic tissue are identified with white arrows. Panels **a–e** taken under white light. Panels **f–j** taken under mCherry filter. **k**, **l** *kmo* disruption phenotypes observed in G2 progeny of **k** 'wildtype' G2 offspring, not carrying the gRNA transgene or **l** G2 offspring carrying the gRNA transgene. Across the experiment, according to treatment (first letter in the treatment code gives the sex of the G0 individuals in that cross and the second letter gives the sex of the transheterozygous G1 individuals in that cross. As an example, "FM" is a treatment where the *vasa*-Cas9 transgene was inherited through a female G0 and its male transheterozygote G1 progeny were then crossed to wildtype females). Raw data is provided in tables below each graph with percentages for each cohort provided in brackets after each raw data value.

also showed that the biased inheritance observed has both a meiotic drive and a homing component. Interestingly, while the vast majority of components tested in that previous *Ae. aegypti* study did not induce homing in the male germline, here we observed higher levels of homing in G1 males than in females. This difference is possibly due to our use of regulatory elements from the *vasa* gene to drive Cas9 expression (known to be a germ-cell marker in both males and females). In contrast, the previous study in *Ae. aegypti* used instead pre-existing Cas9 expressing lines that had been generated to allow germline-transgenesis/mutation−often through a rational choice of genes expressed at high levels in the ovaries.

Given the very high levels of optimization required in *Ae aegypti* to achieve high levels of homing (e.g., >80%)[33], the demonstration of significant homing with the limited components tested is encouraging for further development in *Cx. quinquefasciatus*. Testing of future optimized constructs would greatly benefit from the

development of more efficient transgenesis techniques in *Cx. quinquefasciatus*. Currently, options for this are limited to CRISPR/HDR-based targeted delivery with reports suggesting that traditional insect transgenesis technologies such as *piggyBac* may be non-functional or very inefficient in this species[29]. While successful use of a *Hermes* transposon-based system has been reported in *Cx. quinquefasciatus*[35], this has not been functional in our hands. While HDR-based transgenesis allows for precise integration into the genome, it has been thus far relatively inefficient and it does not allow for the semi-random integration behavior of transposon-based systems—a brute-force method for overcoming potential positional effects of insertion site on transgene expression.

To date, the relatively low homing rates in *Ae aegypti*, even after significant research effort by multiple groups towards their optimization, suggest that homing-based drives in this species may be more appropriately aimed at population-modification (the spreading of disease-refractory transgenes), rather than population suppression. This stands in contrast to *An. gambiae*, where the extremely high levels of homing reported have supported the development of various suppression-drives[22]. If the situation in *Cx. quinquefasciatus* mirrors that of *Ae. aegypti*, then functional modification drives in this species will also require the development of disease-refractory systems, as have been developed for some pathogens in *Ae. aegypti*[36–39]. With transgenic manipulation of *Cx. quinquefasciatus* in its infancy, there is much work to be done here, in identifying not only robust refractory mechanisms (e.g., upregulation of endogenous immune pathways, antiviral RNAi, single-chain antibodies) but also the gene regulatory elements required to direct their spatial and temporal control to disease relevant tissues or developmental stages in the adult female.

Our results showing substantial levels of *kmo*-targeting by Cas9/gRNA RNPs deposition in the eggs by the mother, as well as efficient *kmo*-targeting in the male germline, also support the potential development of alternative genetic control designs such as the development of Toxin-Antidote based gene drive systems (i.e., ClvR/TARE) that may be effective in this species[40,41]. Again, significant optimization of components will be required, such as identifying alternative insertion sites for the *vasa*-Cas9 transgene to increase levels of cutting and egg deposition, and identifying haplosufficient target genes with lethal knockout phenotypes, and engineering their rescue copies. Moreover, our extremely high levels of somatic *kmo*-cutting in trans-heterozygotes of both sexes suggest that this *vasa*-Cas9 insertion may be useful for the development of genetic sterile-insect genetic control design[42]. Finally, other methods of genetic control, for example Y-linked editors[43] or sex-reversal gene drives[44] would greatly benefit from fundamental research elucidating the composition and functional mechanism of the sex-determining region in *Cx. quinquefasciatus*.

In summary, with this work we have extended the list of disease-vectors where engineered CRISPR-based homing gene drives have been demonstrated, to include *Culex* along with *Anopheles* and *Aedes*. This proof-of-principle demonstration is particularly encouraging given the very limited amount of molecular tools and genome editing knowledge available for *Culex* mosquitoes. However, further work is required to bring this initial demonstration to homing efficiencies capable of robustly functioning in the field. We hope that these results will spur research interest in developing the tools and components required to allow the further development of genetic control strategies for *Culex*, and further support the generalizability of these methods in other insects.

## Methods

### Ethics
All the work presented here followed procedures and protocols approved by the UC San Diego Institutional Biosafety Committee, and the Pirbright Institute Biological Agents & Genetic Modification Safety Committee. The work performed at the University of California San Diego complies with all relevant ethical regulations for animal testing and research. Gene-drive experiments were performed in a high-security Arthropod Containment Level 2 (ACL2) barrier facility. All experiments were conducted at The Pirbright Institute IS4L arthropod containment facility under the necessary safety regulations for gene drive research.

### Mosquito rearing and maintenance (UCSD)
Different mosquito lines used in experiments involving the *white*-gRNA6 line originated from a wildtype *Culex quinquefasciatus* (California) strain, a kind gift provided by Anton Cornell (UC Davis), named as the CA-*wt* line. The *Culex vasa*-Cas9 homozygous line used in the split-drive was an HDR-based transgenic line generated previously[26]. The *w4-/w4-* mutant line used for tracking conversion rates in a marked chromosome gene drive was built previously and maintained in the lab for several generations[27]. Mosquitoes were reared at 27 ± 1 °C, 75% humidity, and a 12 h light/dark cycle in the insectary room at the University of California, San Diego. The adults were fed with 10% sugar water. After mating, females were fed with defibrinated chicken blood (Colorado Serum Company, #31142) using the Hemotek blood-feeding system (Hemotek, Blackburn, UK). Egg rafts were collected 4 days after blood feeding. Larvae were fed with fish food floating pellets (Blue Ridge Fish Hatchery, USA). The 3rd instar larvae were later examined and scored with a Leica M165 FC Stereo microscope for fluorescence. All work followed procedures and protocols approved by the Institutional Biosafety Committee from the University of California, San Diego, complying with all relevant ethical regulations for animal testing and research. All maintenance and experiments were performed in a high-security Arthropod Containment Level 2 (ACL2) barrier facility. The wastewater and used containers were disposed of by freezing for 48 hours, and subsequently discarded as biohazardous materials.

### Mosquito rearing and maintenance (TPI)
The *kmo*-gRNA line was generated previously by HDR-based transgenesis, following injection procedures previously published by the Pirbright group[28]. Wild-type individuals used in these experiments originated from the TPRI (Tropical Pesticides Research Institute) strain. Adults were maintained at 27 °C, 70% humidity and 12 h day/night cycle. Egg rafts were collected from adult cages in a 150 ml plastic container filled with 'larval water'. To make this, water that had previously housed L3-L4 larvae was double-filtered through a coffee filter to remove any large organic material (including larvae). Mosquito larvae were raised in deionised $H_2O$ and fed with pelleted pond fish food (Extra Select). Adult mosquitoes were fed *ad libitum* with 10% sucrose solution. A Hemotek blood-feeding system was used to supply defibrinated horse blood (TCS Biosciences, Buckingham, UK) through one layer of hog gut (sausage casing) to adults one week after eclosion.

### Cloning of the *w6*-gRNA homing element construct
The *white*-gRNA6 fragment was synthesized by annealing oligos, and inserted into the double-BbsI restriction site linker of *Cq*-U6-1_2XBbsI-gRNA plasmid (Addgene#169238) to build a *Cq*-U6-1-*w6*-gRNA plasmid. The gRNA scaffold is modified by the insertion of an extra 5 bp loop structure to boost transgenesis as mentioned in our previous studies[26]. The *Cq*-U6-1-*w6*-gRNA fragment was later amplified and cloned into the *white* (CPIJ005542) gene flanked by the homology arms abutting the *w6*-gRNA cutting site. The construct was built by Gibson Assembly using NEBuilder HiFi DNA Assembly Master Mix (New England Biolabs, #E2621). After assembling, the plasmid was transformed into NEB 5-alpha Electrocompetent Competent *E. coli* (New England Biolabs, # C2989), and correct clones were subsequently confirmed by restriction digestion and Sanger sequencing. All primers used to build plasmids generated in this work are listed in Supplementary Table 1.

The plasmid sequence information is available in GenBank, accession numbers are reported in the Data Availability Section.

## Generation of a *w6*-gRNA homing element transgenic line

The injected HDR template (*white*-gRNA6 gRNA-only gene drive) plasmid was prepared using PureLink Expi Endotoxin-Free Maxi Plasmid Purification Kit (Invitrogen, Cat. #A31231), aliquoted based on the concentrations used for the injections, and stored at −80 °C before proceeding to microinjection. All injections were performed on a microinjection station equipped with a FemtoJet 4 microinjector (Eppendorf). A 200 ng/μL of plasmid was injected into the posterior end of embryo eggs of the *vasa*-Cas9 line freshly collected after oviposition (-1 h) to ensure efficient targeting of the germline. The *Culex* egg microinjection is conducted following a previously published protocol[27]. The injected G0s were divided by sex into two different pools, and crossed with wildtype individuals of the opposite sex at a ratio of 3:1–5:1. After mating and blood feeding, egg rafts were collected from each pool and hatched in trays. The 3rd instar larvae of G1 were screened for eGFP fluorescence under a Leica M165 FC stereo microscope. The expression of eGFP and DsRed in the larva indicates the successful integration of the transgene in the Cas9 line. In our study, the initial recovered mosquito carrying the *w6*-gRNA transgene was a male individual and later crossed with ~20 wildtype females to expand the transgenic pool for subsequent gene drive experiments. Nearly all offspring carrying both Cas9/gRNA transgenes were males, except for one female individual carrying the gRNA-eGFP transgene only. The *white*-gRNA6 only transgenic line was built by crossing this single female with multiple *white-gRNA6-only* transgenic males. Verification of the correct insertion was performed by PCRs using primers listed in Supplementary Table 1.

## The split gene drive experiments

Methods for these split gene drive experiments are described in their relevant result sections as well as illustrated in figures. In summary, in the initial *white* gene drive experiments, the inheritance rates of the *w6*-gRNA transgene were calculated as the eGFP+ G2 individuals divided by the total number of G2 offspring. The inheritance rates of the Cas9 transgene were scored as the DsRed+ G2 individuals divided by the total number of G2 offspring. For the subsequent marked chromosome study involving *white*, the conversion rates were scored as the individuals carrying both *w4-* and eGFP+ phenotypic markers divided by the total offspring numbers counted. Similarly, in *kmo* gene drive experiments, the inheritance rates of the *kmo*-gRNA transgene were scored as the *Hr5IE1*-DsRed+ G2 individuals divided by the total number of G2 offspring. The *Hr5IE1*-DsRed can be phenotypically distinguished from the *Opie2*-DsRed marker of the *vasa*-Cas9 carrying transgene, as shown in Supplementary Fig. 3.

## Statistics, reproducibility, and graphical representation

Figures were generated with GraphPad Prism 9, and Adobe Illustrator. Analyses of inheritance ratios were carried out using R version 4.1.3 (R Development Core Team). No statistical method was used to predetermine sample size prior to experiments. Estimated means and 95% confidence intervals were calculated by a Generalized Linear Mixed Model (GLMM), with a binomial ('logit' link) error distribution fitted using the glmmTMB package[45]. Where applicable, initial parameters included the sex of the G1 parent and the G0 grandparent from which the Cas9 transgene had been inherited (*kmo* experiments), otherwise only G1 parental sex was included (*white* experiments), and individual replicates were included as a random effect. The model with the best fit was then chosen by comparing Akaike Information Criterion (AIC) and summarized with 'emmeans', model residuals were checked for violations of assumptions using the 'DHARMa' package[46–48]. No data were excluded from the analyses.

## Reporting summary

Further information on research design is available in the Nature Portfolio Reporting Summary linked to this article.

## Data availability

The plasmid sequences of the constructs generated in this manuscript are available in the GenBank database under accession number NW925705.1 [https://www.ncbi.nlm.nih.gov/nuccore/MW925705.1/], OR459947, and MW417419.1. Additional information is provided in the Supplementary Information. All source data are provided along with this manuscript. They cover the raw phenotypical scoring data collected in the transgenesis and gene drive experiments, which are reported in the Supplementary Data 1–4 files in Microsoft Excel format (.xlsx). All other data and information is available upon request from the authors.

## Code availability

The experimental data and the analyses that support the findings of this study are available in Zenodo with the identifier: https://doi.org/10.5281/zenodo.8005545.

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

## Acknowledgements

We thank the current and previous members of the Gantz and Alphey labs for their support of this project and for valuable comments and edits on the manuscript. The research reported in this manuscript was supported by the University of California, San Diego, Department of Biological Sciences, by the Office of the Director of the National Institutes of Health under award number DP5OD023098 (to V.M.G), by the National Institute of Allergy and Infectious Diseases under the award number R01AI162911 (to V.M.G). National Natural Science Foundation of China grant number 82202559 (to X.F.). L.A. and T.H.S. were funded by the UK Biotechnology and Biological Sciences Research Council [BBS/E/I/00007033, BBS/E/I/00007038, and BBS/E/I/00007039 strategic funding to The Pirbright Institute.

## Author contributions

X.F., V.M.G., T.H.-S., and L.A. conceived the project. X.F., E.M.O., V.M.G., T.H.-S., and L.A. contributed to the design of the experiments. X.F., E.M.O., T.H.-S., D.-K.P., P.T.L., and V.M.G. performed the experiments and contributed to the collection and analysis of data. X.F., E.M.O., T.H-S., L.A., and V.M.G. wrote the manuscript. All authors edited the manuscript.

## Competing interests

V.M.G. is a founder of and has equity interests in Symbol, Inc. and Agragene, Inc., companies that may potentially benefit from the research results described in this manuscript. V.M.G. also serves on both the company's Scientific Advisory Board and the Board of Directors of Synbal, Inc. The terms of this arrangement have been reviewed and approved by the University of California, San Diego in accordance with its conflict of interest policies. L.A. is an advisor to, and has financial or equity interest in, Synvect Inc. and Biocentis Ltd., companies operating in the area of genetic control of pest insects. X.F., T.H.-S., D.-K.P., P.T.L., and E.M.O. declare no competing interests.
