## [Peer Review File · Nature Communications]

Reviewers' Comments:

Reviewer #1:

Remarks to the Author:

In the manuscript "CRISPR-based gene drives generate super-Mendelian inheritance in the disease vector *Culex quinquefasciatus*", Harvey-Samuel et al. provide multiple examples of inheritance bias via CRISPR-Cas9 homing gene drives in the southern house mosquito. While the rate of bias was modest at each locus tested, there was a significant increase in inheritance in a non-Mendelian fashion, paving an important step in future gene drive development for this significant vector of disease. Overall, the experiments show measurable levels of homing and chromosomal inheritance bias, as well as eye mosaicism in kmo gene drive progeny as evidence of RNP and not genetic inheritance of CRISPR-Cas9 elements in some individuals. I recommend acceptance with the following suggestions/modifications:

Line 25 – "demonstrating the possibility of using this technology to control *Culex* mosquitoes"

Given the low increase in biased inheritance, this feels like a strong statement, and perhaps should be removed, as this work seems to provide a path forward for controlling *Culex* mosquitoes using gene drives but does not yet demonstrate this possibility.

Line 169-171 "Additionally, since it seems that the homing process may occur at very low levels (Fig. 2d), homing events may also be happening in the M-to-m condition, although the smaller sample size and the limited resolution of this assay may be hiding this"

This sentence should be reworded to correct grammar ("seems") and made more concise.

Line 193-195 "To better understand the underlying chromosomal conversion events, we designed a follow-up experiment to visualize these events more effectively, that would allow us to confirm homing events independently from other mechanisms of biased inheritance."

I feel it would be more clearly explain the following experiment if this sentence clarifies that the goal of this analysis is to differentiate between homing gene drive events and biased inheritance of gene drive containing chromosomes.

Line 312-314 "Interestingly, while we expected to observe higher levels of mosaicism in the kmo-gRNA G2 progeny than their wildtype siblings, the opposite was the case, although these differences were not statistically assessed."

As this was not statistically assessed this does not feel appropriate in the results section and I would recommend its removal unless this statistical assessment is completed.

Reviewer #2:

Remarks to the Author:

Review of CRISPR-based gene drives generate super-Mendelian inheritance in the disease vector *Culex quinquefasciatus*

Harvey-Samuel and Peng et al.

Harvey-Samuel, Feng and co-authors present their successful efforts to establish a CRISPR-Cas9/Transgene-homing mechanism for gene drive in *Culex quinquefasciatus*. This is an important development: *Culex* species have been challenging to genetically engineer, but are very important disease vectors and control efforts would benefit from gene-drive strategies for disease control. The authors demonstrate a split gene-drive, where Cas9 is expressed from one location and drives the cleavage at another chromosomal location and subsequent homing of a guide RNA transgene. They demonstrate that this leads to an increase of inheritance of the guide RNA encoding transgene at a higher rate than expected to occur by independent segregation of alleles. Their clever use of genetic and visual markers and strategic mosquito crossings/screenings allow them to quantify

both an effect of the homing and a bias of inheritance of the transgene over wild type chromosome. The test drives here show more modest inheritance increases than has currently been achieved in *Anopheles* and *Aedes* species, but this is a pivotal (and challenging) starting point to building gene-drive mechanisms with improved efficiencies.

It is my opinion that this manuscript and the work described is exceptionally clear and thorough and describe an important advancement in insect molecular biology and genetic engineering. The figures are well constructed and are a great support to both the methods and results text. I saw no deficiency in data analysis or interpretation; the molecular work and genetics methodology is sound and the conclusions are valid. The statistical tests seem appropriate, but I am not familiar enough with the nuances of the statistics described to be highly critical in that regard.

In terms of reproducibility, I raise a few easily addressed concerns:

1. The authors state that the plasmid sequences will be published on GenBank, but it is important for reproducibility and translation of this work that these sequences are well annotated on GenBank as well. This may well be the intention, but is important enough to mention here.
2. In the methods section on transgenesis, how are the embryos treated after injection until hatching? I believe this varies between research groups and has an important impact on hatching. It should be described here or an open access, published protocol referenced.

Additionally, I have a few minor suggestions for improvement of the manuscript:

3. In the third paragraph of the discussion, I believe the text comparing the promoters used here versus in *Aedes*, could be clarified to indicate that *vasa* is highly expressed also in the male germline. The implication, as I understand it is that the *Aedes* work used promoters that drive expression very highly in the ovaries, but not the testes, which could explain the lack of homing in the male germline in *Aedes*.
4. The authors need to identify figure 4 in section starting on line 255.
5. In the methods section on transgenesis, how are the embryos treated after injection until hatching? I believe this varies between groups and has an important impact on hatching. It should be described here or a published protocol referenced.
6. The explanation of the M and F code used in Figure 4 isn't explained until the legend in Figure 5 and is not intuitive. The explanation should be added to the Figure 4 legend as well.

Authors' responses to the reviewers' comments

We thank both reviewers for their feedback on the manuscript and recognition of this work in paving an important step for the development of gene drive systems in *Culex* mosquitoes. We are providing a revised version of the manuscript, including the suggested edits and revisions, with comments highlighting the edits addressing each specific numbered comment. Please find below point-by-point responses in blue text.

Reviewer #1 (Remarks to the Author):

In the manuscript “CRISPR-based gene drives generate super-Mendelian inheritance in the disease vector *Culex quinquefasciatus*”, Harvey-Samuel et al. provide multiple examples of inheritance bias via CRISPR-Cas9 homing gene drives in the southern house mosquito. While the rate of bias was modest at each locus tested, there was a significant increase in inheritance in a non-Mendelian fashion, paving an important step in future gene drive development for this significant vector of disease. Overall, the experiments show measurable levels of homing and chromosomal inheritance bias, as well as eye mosaicism in kmo gene drive progeny as evidence of RNP and not genetic inheritance of CRISPR-Cas9 elements in some individuals. I recommend acceptance with the following suggestions/modifications:

1. Line 25 – “demonstrating the possibility of using this technology to control *Culex* mosquitoes”

Given the low increase in biased inheritance, this feels like a strong statement, and perhaps should be removed, as this work seems to provide a path forward for controlling *Culex* mosquitoes using gene drives but does not yet demonstrate this possibility.

Agreed. We have deleted this sentence in the abstract.

2. Line 169-171 “Additionally, since it seem that the homing process may occur at very low levels (Fig. 2d), homing events may also be happening in the M-to-m condition, although the smaller sample size and the limited resolution of this assay may be hiding this”. This sentence should be reworded to correct grammar (“seems”) and made more concise.

Agreed. We have modified this sentence as “Additionally, since it seems that the homing process may occur at very low levels (Fig. 2d), homing events may also be happening in the M-to-m condition, which might be obscured with the smaller sample size and the limited resolution of this assay”.

3. Line 193-195 “To better understand the underlying chromosomal conversion events, we designed a follow-up experiment to visualize these events more effectively, that would allow us to confirm homing events independently from other mechanisms of biased inheritance.” I feel it would be more clearly explain the following experiment if this

sentence clarifies that the goal of this analysis is to differentiate between homing gene drive events and biased inheritance of gene drive containing chromosomes.

Agreed. We have modified this passage as suggested.

4. Line 312-314 “Interestingly, while we expected to observe higher levels of mosaicism in the kmo-gRNA G2 progeny than their wildtype siblings, the opposite was the case, although these differences were not statistically assessed.” As this was not statistically assessed this does not feel appropriate in the results section and I would recommend its removal unless this statistical assessment is completed.

Agreed. This passage has been removed.

Reviewer #2 (Remarks to the Author):

Review of CRISPR-based gene drives generate super-Mendelian inheritance in the disease vector *Culex quinquefasciatus*

Harvey-Samuel and Peng et al.

Harvey-Samuel, Feng and co-authors present their successful efforts to establish a CRISPR-Cas9/Transgene-homing mechanism for gene drive in *Culex quinquefasciatus*. This is an important development: *Culex* species have been challenging to genetically engineer, but are very important disease vectors and control efforts would benefit from gene-drive strategies for disease control. The authors demonstrate a split gene-drive, where Cas9 is expressed from one location and drives the cleavage at another chromosomal location and subsequent homing of a guide RNA transgene. The demonstrate that this leads to an increase of inheritance of the guide RNA encoding transgene at a higher rate than expected to occur by independent segregation of alleles. Their clever use of genetic and visual markers and strategic mosquito crossings/screenings allow them to quantify both an effect of the homing and a bias of inheritance of the transgene over wild type chromosome. The test drives here show more modest inheritance increases than has currently been achieved in *Anopheles* and *Aedes* species, but this is a pivotal (and challenging) starting point to building gene-drive mechanisms with improved efficiencies.

It is my opinion that this manuscript and the work described is exceptionally clear and thorough and describe an important advancement in insect molecular biology and genetic engineering. The figures are well constructed and are a great support to both the methods and results text. I saw no deficiency in data analysis or interpretation; the molecular work and genetics methodology is sound and the conclusions are valid. The statistical tests seem appropriate, but I

am not familiar enough with the nuances of the statistics described to be highly critical in that regard.

In terms of reproducibility, I raise a few easily addressed concerns:

1. The authors state that the plasmid sequences will be published on GenBank, but it is important for reproducibility and translation of this work that these sequences are well annotated on GenBank as well. This may well be the intention, but is important enough to mention here.

The Cas9 and *kmo*-gRNA plasmids used in this study are generated in a previous study (Feng et al., 2021) with Genbank accession number ID: MW925705 and MW417419, respectively. We have uploaded the plasmid of *white6*-gRNA in the Genbank with the accession number OR459947. All accession numbers have been listed in the Data Availability Section.

2. In the methods section on transgenesis, how are the embryos treated after injection until hatching? I believe this varies between research groups and has an important impact on hatching. It should be described here or an open access, published protocol referenced.

A detailed *Culex*-egg microinjection protocol from each group was previously published and the text has been modified to better direct the reader to each of such publications.

Additionally, I have a few minor suggestions for improvement of the manuscript:

3. In the third paragraph of the discussion, I believe the text comparing the promoters used here versus in *Aedes*, could be clarified to indicate that *vasa* is highly expressed also in the male germline. The implication, as I understand it is that the *Aedes* work used promoters that drive expression very highly in the ovaries, but not the testes, which could explain the lack of homing in the male germline in *Aedes*.

Agreed. We have modified this paragraph to include a sentence noting the male and female germline expression of *vasa*, in contrast to the previous study in *Aedes aegypti* where only high levels of expression in the ovaries were used as a filter for choosing regulatory elements to drive Cas9.

4. The authors need to identify figure 4 in section starting on line 255.

Yes! Thanks for catching the missing reference to Figure 4.

5. In the methods section on transgenesis, how are the embryos treated after injection until hatching? I believe this varies between groups and has an important impact on hatching. It should be described here or a published protocol referenced.

Please see our response in Comment #2.

6. The explanation of the M and F code used in Figure 4 isn't explained until the legend in Figure 5 and is not intuitive. The explanation should be added to the Figure 4 legend as well.

We have updated Figure 4 with the "M" and "F" codes in the legend.